# Glycolysis of Polyurethanes Composites Containing Nanosilica

**DOI:** 10.3390/polym13091418

**Published:** 2021-04-27

**Authors:** Jesus del Amo, Ana Maria Borreguero, Maria Jesus Ramos, Juan Francisco Rodríguez

**Affiliations:** Department of Chemical Engineering, Institute of Chemical and Environmental Technology, ITQUIMA, Avda, Camilo José Cela s/n, University of Castilla-La Mancha, 13004 Ciudad Real, Spain; jesus.delamo@uclm.es (J.d.A.); anamaria.borreguero@uclm.es (A.M.B.); mariajesus.ramos@uclm.es (M.J.R.)

**Keywords:** glycolysis, rigid polyurethane foams, nanosilica, crude glycerol, polyurethane composites, mechanical properties, thermal properties

## Abstract

Rigid polyurethane (RPU) foams have been successfully glycolyzed by using diethylene glycol (DEG) and crude glycerol (CG) as transesterification agents. However, DEG did not allow to achieve a split-phase process, obtaining a product with low polyol purity (61.7 wt %). On contrary, CG allowed to achieve a split-phase glycolysis improving the recovered polyol purity (76.5%). This is an important novelty since, up to now, RPUs were glycolyzed in single-phase processes giving products of low polyol concentration, which reduced the further applications. Moreover, the nanosilica used as filler of the glycolyzed foams was recovered completely pure. The recovered polyol successfully replaced up to 60% of the raw polyol in the synthesis of RPU foams and including the recovered nanosilica in the same concentration than in glycolyzed foam. Thus, the feasibility of the chemical recycling of this type of polyurethane composites has been demonstrated. Additionally, PU foams were synthesized employing fresh nanosilica to evaluate whether the recovered nanosilica has any influence on the RPU foam properties. These foams were characterized structurally, mechanically and thermally with the aim of proving that they met the specifications of commercial foams. Finally, the feasibility of recovering the of CG by vacuum distillation has been demonstrated.

## 1. Introduction

Polyurethanes (PUs) are the sixth most used polymers group all over the world with a production of 22.3 million tons per year [1,2,3,4]. The PUs compounds are generally thermoset polymers whose synthesis consists in the reaction of nucleophilic addition between a multifunctional alcohol (polyol) and a di- or tri-isocyanate, resulting in a reticular urethane structure [4,5,6], as shown in reaction (1).
$$\begin{array}{ccc} {\left(HO-R-OH\right)}_{n}+(O=C= & N-R^{\prime }-N=C=O{)}_{n}\to [-CO-NH- & R^{\prime }-NH-CO-O-R-O{]}_{n}\;\left(1\right)\\ \mathrm{Polyol} & \mathrm{Isocyanate} & \mathrm{Urethane} \end{array}$$

The isocyanates employed are classified according to the chain structure, finding aromatic derivatives (methyldiphenylisocyanate, MDI, and toluendiisocyanate, TDI) or aliphatic, employing those from aromatic compounds in the 90% of polyurethane production [7].

In the synthesis of PUs, together with the polyol and the isocyanate, a variety of additives as catalysts, surfactants, antioxidants, pigments and fillers are used with the aim of adapting the properties of the product to its further application [8].

PUs find a wide range of applications and can be classified mainly in foams and in the so-called CASEs (coatings, adhesives, sealants and elastomers). In addition, foams can be subdivided into flexible, such as those used in mattresses, comfort, packaging and car seats and rigid, commonly applied in building and in all type of refrigeration devices insulation. Due to the multitude of applications commented, an increasing amount of non-biodegradable PU wastes are generated worldwide. Traditionally, polyurethane wastes have found their end of life destination in landfills, but the increase of waste generation, the rise in PU raw materials prices, the space restrictions in the landfills, the new environmental regulations and the pressure of public opinion are forcing to find a greener destiny for PU wastes [5,9]. Therefore, research to get a solution for this environmental issue is an urgent task, to convert the PU production and application into a circular economy activity.

On other hand, the general trend in the polyurethane market is the expansion of more advanced materials with improved properties. One of the most frequently used way to modify the physical properties of PUs is the incorporation of fillers of different chemical nature. In the case of rigid polyurethane foams, the incorporation of nanosilica particles with the aim of improving their mechanical properties with respect to conventional ones is one of the most employed. Specifically, properties such as the resistance to traction, the resistance to erosion, the insulation ability and the resistance to thermal degradation can be improved adding nanosilica [10,11,12,13,14]. For this reason, this kind of advanced materials are gaining market share every day. The growing success of such composites make urgent to find solutions for their recycling, specifically intended to recover the filler. Efforts for a feasible method of recycling the raw materials used in the synthesis of conventional PUs have been focused on the chemical recycling, which is a method that allows one to deploy the crosslinking structure of the PUs yielding the starting polyol and several isocyanate related products. Thus, they should be also the best alternative for the recycling of PU based composites [15].

Chemical processes allow one to recover the starting polyol that can be reused replacing partially or totally the fresh one. Between the chemical processes (hydrolysis, aminolysis, etc.), the most important one is the glycolysis, because it provides the best outcomes with respect to the quality of the recovered product at mild reaction conditions [5]. The glycolysis process is based on a transesterification reaction, in which the high molecular weight polyol attached to the isocyanate part of the urethane is exchanged by the short chain glycol [5,16]. A general scheme of the PU glycolysis is shown below (Scheme 1).

It is also important to point out that the use of a large excess of glycol, immiscible with the polyols used in flexible foams, allows one to obtain a biphasic product, in which the upper phase is mostly composed by the recovered polyol, while the bottom phase contains the excess of glycolysis agent and reaction byproducts. Thus, the recovered polyol has higher quality than the product from a single-phase process, increasing the possibilities of further application. However, according to the literature review, the glycolysis of rigid polyurethane foams has been uniquely conducted throughout single-phase processes; due to the similar polarity of the polyols employed for the synthesis of these foams with the common glycolysis agents [3]. In this work, the use of crude glycerol and a higher amount of the SnOct catalyst allowed one to reach for the first time a split-phase process achieving a high polyol purity (77%). On the other hand, there is no previous works related to the glycolysis of polyurethane composites containing inorganic fillers, as it is reported in this work, where both, the polyol and the filler (nanosilica) have been successfully reused in the synthesis of new rigid PU with suitable properties for commercial applications.

## 2. Experimental

The sequence of the performed experiments carried out in this work is shown in Scheme 2. The materials, experimental setups and methodologies employed are detailed in the next subsections.

### 2.1. Materials

#### 2.1.1. Glycolysis

Glycolysis agents are composed of glycol and catalyst. The glycols employed were diethylene glycol (purity 99.8%, supplied by Campi y Jové S.A., Barcelona, Spain) and crude glycerol (purity 80%, provided by Biocombustibles de Cuenca, Cuenca, Spain). Stannous octoate (purity 95%, provided by Sigma-Aldrich, Madrid, Spain) was added as a catalyst.

#### 2.1.2. Polyurethane Synthesis

Rigid PU foams were synthesized from polymeric methylene diphenyl diisocyanate (PMDI) and a raw rigid polyol (Mn = 555 g/mol, OH = 455 mg KOH/g, acidity = 0.15 mg KOH/g, viscosity = 5250 cp and humidity = 0.1%). Different percentages of the raw rigid polyol were replaced with the recovered polyols obtained from the glycolysis of the PU containing nanosilica as a filler. Nanosilica fresh or recovered was added to the PU foam synthesis. PMDI was supplied by Poliuretanos Aismar, S.A (NCO content = 31%). The catalyst used was Tegoamin BDE and the surfactant was Tegostab B8404, both supplied by Evonik Degussa International AG. Deionized water was used as a blowing agent. The recipes of the different synthesized foams are shown in Table 1. The glycolyzed foams were synthesized with the proportions of column P100/R0.

### 2.2. Glycolysis Process

Glycolysis reactions of rigid polyurethane foams with nanosilica as a filler were performed at a laboratory scale. The installation consisted in a jacketed 2 L flask, heated with silicone oil derived from a circulation thermostat. The reactor was thermally insulated to reduce heat losses. At the top there was a refluxing condenser, N_2_ intake to keep inert atmosphere and to prevent oxidation, a mouth for feeding the glycolysis agent and a stirring head with a range of 40–2000 rpm, which drove an agitator Rushton type of 6-blades. Sampling during the reaction was carried out by means of a 25 mL pipette through one of the mouths of the reactor lid and, for the discharge of the product, there was a valve at the bottom of the reactor vessel. The polyurethane foam used in the present investigation was cut in arbitrary particle size distribution pieces with values between 5 and 25 mm in diameter. The installation was placed in a showcase with a gas extraction system. The experimental procedure during the glycolysis reactions comprised the assembly of the reflux condenser and stirring system, subsequent charge of the reactor with the glycolysis agent (glycol and catalyst) and the start of heating and vigorous stirring (300 rpm) until reaching the operation temperature, working always under inert atmosphere. The catalyst concentration in the glycolysis agent (GA) was kept constant at 2.3 wt % and as glycol were employed diethylene glycol or crude glycerol. The used PU:GA ratios varied from 1:1.5 to 1:2, with the aim of using an excess of the glycolysis agent with respect to the PU foam that allowed one to obtain a product with split phase. Once the reaction temperature was reached (190 °C), the foam was fed continuously, completing this step in one hour. After six hours of reaction, the final product was extracted by the discharge valve of the reactor and poured in a funnel to separate the different phases.

Scheme of the PU glycolysis employing DEG (a) or CG (b) as a glycolysis agent (Scheme 3).

### 2.3. Polyurethane Synthesis Process

The foam formation process was carried out in different stages, the first one consisted of mixing the polyol, the nanosilica, the surfactant, the water and the amine catalyst, using a stirring rate of 1000 rpm. Once the mixture was perfectly homogeneous, the stirring speed was increased up to 2000 rpm; then, the appropriate amount of isocyanate (PMDI) was added to the mixture keeping the stirring for 5 s, moment in which the foam began to grow. Finally, the foams were cured at room temperature for 24 h.

### 2.4. Characterization Techniques

#### 2.4.1. Molecular Weight and Product Composition Determination by Gel Permeation Chromatography (GPC)

To carry out the determination of the molecular weight and the distribution of molecular weights, it was employed a Viscotek GPCmax VE-2001 TDA 302 Detectors chromatograph with two peristaltic pumps, automatic injection system, rack for 120 vials, electric oven, two columns Water Styragel Column HR 2 (pore size 500 Å, molecular weight 0–100 g/mol) and HR 0.5 (pore size 50 Å, molecular weight 500–20,000 g/mol) and triple detection, consisting of a LALS (low angle light scattering) detector, a RALS (right angle light scattering) detector and viscosity detector. The GPC equipment was supplied by Malvern Products Iesmat company (Madrid, Spain) and the columns by Waters Cromatografía, S.A. (Barcelona, Spain). The set of columns allowed the detection of molecular weights in the range of 10–20,000 g/mol, suitable for all synthesized products. The equipment was controlled by a computer system provided with the OmniSEC 4.5.6 program that recorded and analyzed the results. The test conditions were a temperature of 40 °C, a flow rate of 1 mL/min, a sample concentration of 10 mg/mL and an injection volume of 50 µL. The area of a peak was related to the concentration of the substance in the sample by means of the response factors, which were obtained by calibrating with samples of pure substances. 

#### 2.4.2. Structural Studies Using Fourier Transform Infrared Spectroscopy (FTIR)

The characterization of the obtained products has been carried out using a Varian 640-IR FT-IR spectrophotometer in the range of 4000–400 cm^−1^, 8.0 cm^−1^ resolution and 16 scans, with a program called Varian Resolution Pro Software, version 5.0.

#### 2.4.3. Measurement of Hydroxyl Index (iOH)

The polyols hydroxyl number was determined by a standard titration method (AOCS Official Method Cd 13-60) [17]. 

First, a solution of acetic anhydride in pyridine is used for the esterification of the hydroxyl groups. Then, the excess of anhydride is titrated with a standard solution of sodium hydroxide KOH 0.5 M. Finally, Equation (1) is applied to calculate the hydroxyl number.
(1)$$Hydroxyl\;number=\frac{\left[B+\left(\frac{W\;A}{C}\right)-S\right]\times M\times 56.1}{W}$$
where *S*, *A* and *B* are the volume (in mL) of KOH spent during the titration of the sample for first and second procedures and the reference sample, respectively; *W* and *C* are the sample weight in grams for the first and second procedures, respectively, and *M* the molarity of the KOH solution.

### 2.5. Nanosilica Shape and Size Characterization

#### 2.5.1. Dynamic Light Scattering (DLS)

The equipment Zetasizer Nano ZS from Malvern Instruments permitted the evaluation of the fresh and recovered nanosilica to obtain the particle size distribution in volume and number dispersed in distilled water by DLS.

#### 2.5.2. Scanning Electron Microscopy (SEM)

The shape and size of the of the fresh and recovered nanosilica were characterized by SEM, using Quanta 250 equipment (FEI Company) with a tungsten filament operating at a working potential of 15 or 20 kV and magnification of ×50,000.

### 2.6. Foams Characterization

#### 2.6.1. Apparent Density

Foams apparent density was measured according to ISO 845 standard. The density tests were conducted on rectangular shape foam samples with the dimensions of 51 mm × 51 mm × 26 mm. Then, the specimens were weighed to calculate the apparent density as the ratio between the mass and volume. 

#### 2.6.2. Compression Test

Compressive properties of foams were measured following the standard ASTM D1621. Uniaxial compression tests were performed using an MTS 370.02 testing instrument, at a crosshead speed of 2.5 mm/min. The foam specimens shape and size were prism of 5.1 cm × 5.1 cm × 2.6 cm and each sample was tested 3 times. From these test results, the Young’s modulus (E) was determined as the slope value of the initial part of the compression curves, while the maximum compressive strength (σ_max_) was determined as the point in which the plateau region starts or, in the absence of a plateau, in the inflection point of the curve. 

#### 2.6.3. Foam Cell Structure (SEM)

The cell structure of the synthesized rigid PU foams was characterized by SEM, using Quanta 250 equipment (FEI Company) with a tungsten filament operating at a working potential of 12.5 or 15 kV, in order to study the influence of the amount of the recovered polyol and of the presence of nanosilica fresh or recovered in the cell size and structure of the foams. The average foam cell size was measured by the Motic Image Plus program.

#### 2.6.4. Effective Conductivity

The thermal behavior of the synthesized rigid foams was studied by means of a homemade equipment described and proved in previous works [18,19]. The experimental device and the temperature and heat flux sensors positions are shown in Figure 1. 

Tests were carried out applying two thermostatic bath set-point step changes from 18 to 40 and from 40 to 18 ± 0.1 °C while temperatures at different foam sample locations and the inlet and outlet heat fluxes were registered with time. The foams samples dimensions were of 3 × 6 × 10 cm^3^. Six thermocouples of K-type were located across the probe thickness: two in the external sample surface (*T_up_*), two in the middle (*T_middle_*) and the other two on the aluminum cell (*T_down_*). The heat fluxes were measured by using heat flow sensors PU22T for monitoring the inlet (*Q_in_*) and outlet (*Q_out i_*) heat fluxes online (Figure 1b).

From the measured signals, the effective thermal conductivity at the final steady state (*k*) was quantified using Equation (2), where *Q_in_* is approximately equal to *Q_out2_*.
(2)$$k=\frac{Q_{in}\cdot x_{f}}{(T_{down}-T_{up)}}$$
where *Q_in_* is the inlet heat flux at the final steady state condition (W/m^2^) and *χ_f_* (m) is the foam thickness.

### 2.7. Distillation Process

The distillation process was carried out with a Kuhelrohr Short Path Distillation (Aldrich chemistry) that allows operating up to temperatures of 220 °C. It is equipped with a vacuum pump TELSTAR RD-9 with a nominal flow of 9 m^3^/h and a monophasic motor of 1.1 kW, a vacuum trap and a vacuum meter and controller DIVATRONIC DT1. This short-path vacuum distiller is specially designed to separate high boiling point compounds (normally, around 300 °C).

## 3. Results and Discussion

### 3.1. Feasibility Study of the Glycolysis Process of Rigid PU Foams with Nanosilica as a Filler

Several glycolysis reactions of rigid polyurethane foam containing nanosilica as a filler were carried out. The reaction temperature, stirrer type and stirring speed were the optimal ones for flexible polyurethane foams determined in previous works [20,21,22,23]. However, the catalyst concentration was slightly increased to 2.3% in order to achieve complete degradation of polyurethane and a split phase reaction system. Besides, in order to improve the phase splitting, different ratios of PU foam to glycolysis agent were assayed. The reaction recipes for the different PU foam to glycolysis agent ratios appear in Table 1.

After six hours of reaction with DEG, the products obtained presented just one phase even for the case of a ratio 1:2, with a maximum polyol purity of 61.7 wt %, concentration obtained by means of GPC analysis [24]. Figure 2 shows the GPC analysis of the glycolysis product obtained using DEG and a ratio of PU: G.A. of 1:2 compared to those from the raw polyether polyol and DEG.

Besides, FTIR analysis were carried out to complete the characterization of the glycolysis product obtained. Figure 3 shows these analyses in comparison with the infrared of the raw polyol used in the synthesis of the glycolyzed foams.

The FTIR spectra of raw polyol and the glycolysis product presented the typical functional groups of a polyol, as the signals corresponding to the hydroxyl groups OH at 3460 cm^−1^, CO groups associated with hydroxyl groups at 1000–1400 cm^−1^, CH bonds of the aliphatic carbons of the chain polyol at 2800–3000 cm^−1^ and methylene groups at 1452 cm^−1^ [25]. In the case of the glycolysis product, the signal from the hydroxyl groups had a higher intensity than in the crude polyol due to the presence of the diethylene glycol employed in the reaction. Besides, the glycolysis product present two additional small signals corresponding to the reaction byproducts of the glycolysis, these signals correspond to the carbamates groups produced by C = O signal (1736 cm^−1^) and the amine groups produced by the NH_2_ signal (1625 cm^−1^) [26].

Thus, it was not possible to get a split phase reaction system in the range of conditions assayed with very few possibilities of finding such behavior out of this range. As it was pointed out previously, the higher polarity of the recovered polyol used to make the rigid foam, compared to those employed for the synthesis of the flexible ones, allows its solubility in the glycol. Therefore, in the case of using DEG in the glycolysis agent, it is possible to recycle the rigid PU foams containing nanosilica but obtaining just a single phase product restricting the range of further applications.

Looking for a glycol with higher polarity and commercial availability, in order to get a biphasic product, the crude glycerol obtained as a byproduct from biodiesel production was used. Glycerol has already demonstrated its applicability as a glycolysis agent [27]. The assayed PU foam to the glycolysis agent ratio was 1:1.5, with the general recipe shown in Table 1.

In this case, it was possible to obtain a split phase product. Additionally, to the conventional liquid upper and bottom phases, in the bottom of the funnel appeared a solid phase, corresponding to the nanosilica employed as filler in the polyurethane foams glycolyzed.

Figure 4 shows the chromatograms of the upper and bottom phases (UP and BP, respectively) at the end of reaction, 360 min, together with GPC chromatograms of raw rigid polyether.

In the upper phase, peaks I–IV correspond to the recovered polyol, because the retention times of the peaks coincide with those of the raw polyether polyol employed in the synthesis of the nanosilica PU foams. It can be also seen that the crude glycerol is partially soluble in the UP, since peak VI corresponds to it.

In the bottom phase, very slight losses of polyol are observed, being this phase mainly composed of crude glycerol (Peak VI) and reaction byproducts (Peak V).

Therefore, it has been demonstrated that it is possible to carry out the glycolysis process of rigid polyurethane foams containing nanosilica as filler using crude glycerol as degrading agent obtaining a split phase product, with similar reaction conditions optimized in previous works for flexible PU foams [20,21,22,23].

#### 3.1.1. Characterization of Upper and Bottom Phases

In order to study the purity of the recovered polyol, the different concentration in wt % was obtained by means of GPC analysis [24]. The results are shown in Table 2. 

The upper phase had a concentration of 76.5 wt % by weight in recovered polyol, impurified with a 23.5 wt % of crude glycerol. Thus, this phase could be used as partial replacement of the raw polyol in the synthesis of new polyurethane foams.

Figure 5 shows the infrared spectra of the commercial polyol used in the synthesis of the glycolyzed foams and of the UP and BP of the obtained product from glycolysis.

The FTIR spectra of raw polyol and UP present the typical functional groups of a polyol, showing the signals corresponding to the hydroxyl groups OH at 3460 cm^−1^, CO groups associated with hydroxyl groups at 1000-1400 cm^−1^, CH bonds of the aliphatic carbons of the chain polyol at 2800-3000 cm^−1^ and methylene groups at 1452 cm^−1^ [25].

Furthermore, the FTIR spectra correspond to the UP presented two additional small signals corresponding to the reaction byproducts of the glycolysis, these signals correspond to the C = O groups (1736 cm^−1^) produced by the transesterification carbamates and the amine groups produced by NH_2_ signal (1625 cm^−1^) correspond to products similar to toluenediamine [26]. Besides, these reaction byproducts have a molecular weight of approximately 360 g/mol according to GPC results (Table 2). The signals corresponding to the reaction byproducts presented lower intensity compared to the rest, since they are in low concentration, in fact, they were not detected by GPC. Thus, these results corroborated the low concentration of the reaction byproducts in the upper phase. Another important result was that the signal intensity of the hydroxyl groups in the recovered product was higher than in the raw polyol, which indicated a higher concentration of functional alcohol groups in the recovered product, due to the presence of crude glycerol.

The characteristics of this upper phase or recovered polyol are like those of other recovered polyols and allow foreseeing a good result when foaming it, although adjusting the isocyanate index due to the increase of hydroxyl groups [28].

The FTIR spectra of the bottom phase spectra shows the same peak structure than the upper phase, but the intensity of the OH group was greater than in the upper phase, and this was due to the OH bonds presented in the crude glycerol, principal component in the bottom phase, supporting the results obtained by GPC.

#### 3.1.2. Characterization of Solid Phase

One of the main objectives of this research was to recover, characterize and reuse the nanosilica contained as a filler in the residual polyurethane foams.

The solid phase from the glycolysis product was washed with acetone, filtrated under vacuum and dried at 80 °C. Figure 6 shows the FTIR spectra of fresh and recovered nanosilica.

The FTIR spectra of fresh and the recovered filler are practically identical, presenting signals at wavelengths corresponding to Si-O-Si bonds (1050 and 808 cm^−1^) [29]. Therefore, according to these results the solid phase is nanosilica of high purity, which should be suitable to be reused as filler in the synthesis of new RPU foams.

Additionally, XRD analysis were carried out to demonstrate the high quality of the recovered product. Figure 7 shows the XRD analyzes of fresh and recovered nanosilica samples.

From these results it is possible to conclude a high quality of the recovered nanosilica, since both fresh and recovered have the same spectra, and other XDR analysis of nanosilica found in the literature [30,31]. Besides, it can be seen that a broad peak was obtained (2θ varied from 13 to 35°), which agree with the fact that the silica was in nanosize. 

To verify the nanosize of the recovered nanosilica, DLS analysis were carried out for both, the fresh and the recovered nanosilica. Figure 8 shows these analysis in % number (a) and in % volume (b).

From these results it can be verified that both nanosilicas were in the range of nanosize materials, resulting in an average value in number of 294.9 and 139.8 nm for fresh and recovered nanosilica, respectively; and in volume of 296.9 and 145.1 nm for fresh and recovered nanosilica, respectively.

Additionally, SEM pictures of the nanomaterials are shown in Figure 9.

As can be seen in Figure 9, the nanoSiO_2_ preserved the spherical shape after the glycolysis process and particles with sizes around the average values obtained by DLS were observed.

### 3.2. Synthesis of Rigid PU Foams Using Recovered Polyol and Filler

The main objective of the glycolysis process consists in the recovery of the reagents used in the foam synthesis to reuse them in the production of new PU foams. 

The main polyol properties regarding the foaming process are hydroxyl number, functionality and average molecular weight. The hydroxyl number was measured as described previously (Section 2.4.3). The average molecular weight was calculated from GPC analysis and the functionality using Equation (3).
(3)$$f=\frac{OH\cdot Mn}{56,100}$$

Table 3 summarizes the values of these properties for the recovered polyol in comparison to those typical from commercial raw rigid polyether polyols.

The properties of the recovered polyol were in the typical range of the commercial ones, therefore, the upper phase presented suitable characteristics for being foamed.

The quantity of isocyanate required (Table 4) was adjusted from the hydroxyl number value of the recovered polyol according to the procedure described in a previous work, looking for an isocyanate index of 106 [22].

Foams were synthesized with recovered polyol as a replacement of the fresh one up to a 75 wt %. In addition, fresh and recovered nanosilica were incorporated in a 2.3 wt %, in order to see if the recovered nanosilica promoted similar modification of the PU foams properties than the fresh one. The selected concentration of nanosilica is similar to that used in previous studies [10,11,12,13,14].

Table 4 shows in detail the assayed recipes using the recovered polyol from rigid PU foam scraps with nanosilica as a filler.

The rigid PU foams synthesized grew up properly up to a 60 wt % of raw polyol replacement by the recovered one, while a content of a 75 wt % of recovered polyol caused a great height decrease. Thus, the main properties (apparent density, maximum compressive strength, Young’s modulus and thermal conductivity) of the foams containing up to a 60 wt % of the recovered polyol will be tested in the following sections. 

### 3.3. Characterization of the Synthesized Rigid PU Foams: Influence of the Recovered Polyol and Filler

#### 3.3.1. Apparent Density

Table 5 summarizes the apparent densities of the rigid PU foams synthesized with different contents of recovered polyol and employing nanosilica fresh or recovered. 

According to Table 5 results, the rigid PU foam synthesized exclusively from the commercial polyol presented a density value around 50 kg/m^3^. Introducing recovered polyol in the foam recipe reduced the molecular weight of the polyols mixture (Mn raw polyol = 555 g/mol; Mn of recovered polyol = 460.11 g/mol), promoting higher densities. This behavior agrees with those observed by several authors [32,33,34]. This effect could also be justified considering the increase of the foaming system viscosity, which hampers the CO_2_ expandability, resulting in foams with slightly lower heights and, therefore, with higher densities [32,33,34]. To test this hypothesis, the viscosities of the pure polyol and of the different mixtures of pure and recovered polyol employed in the synthesis of the foams were measured. Table 6 shows these results, observing that the higher the percentages of polyol recovered, the higher the viscosity.

On the other hand, the density of the foam synthesized with fresh nanosilica and 0.0 wt % of recovered polyol was also measured, demonstrating that using fresh or recovered nanosilica did not significantly influence on this property, since their densities were 49.65 and 48.15 kg/m^3^, respectively. Finally, considering that the foam without nanosilica presented a density value of 45.61 kg/m^3^, it can be also stated that this filler increased the foams final density. This effect agrees with that observed for other fillers, that of increased the viscosity of the foaming system, hampering the CO_2_ expansion and obtaining denser foams [35,36,37].

#### 3.3.2. Compression Test

In order to check the suitability of the synthesized foams for commercial application, compression tests were carried out. 

The compression tests of the synthesized polyurethane foams were carried out as described in Section 2.5.2., and their maximum compressive strength and the Young’s modulus were determined (Table 7).

It can be observed from Table 7 results that the maximum compressive strength and Young’s modulus values were practically the same for a pure polyol replacement up to a 30 wt % and with just variations of 7% and 28% for the σ and E, respectively, when the recovered polyol supposes a 60 wt % of the total polyol. Therefore, it can be concluded the use of recovered polyol as a replacement of the pure polyol up to percentages of 60 wt % did not significantly influence the mechanical properties. In addition, the σ and E for the foam without nanosilica were 469.78 kPa and 11.60 MPa, respectively, while they increased to 537.01 kPa and 13.61 MPa for the case of using fresh nanosilica and to 510.34 kPa and 13.50 MPa when using the recovered one. Thus, the improvement of the mechanical properties using nanosilica as a filler has been proved, employing fresh or recovered nanosilica, without presenting a significant influence on the mechanical properties if the nanosilica was fresh or recovered.

Finally, it is worthy to highlight that the use of the recovered products, both polyol (up to a 60%) and nanosilica, did not influence significantly on the maximum compressive strength, demonstrating the good quality of the recovered products. Furthermore, according to the bibliography, the rigid polyurethane foams with Young’s modulus values between 7 and 19 MPa are suitable for their main applications [38]. Therefore, the foams with up to a 60 wt % of recovered polyol and nanosilica in the present investigation could be used in their common applications.

#### 3.3.3. PU Foams Thermal Characterization

The effective thermal conductivity, k, of the foams is also one of their most important properties since they are applied for building insulation and commercial refrigeration.

Table 8 shows the effective thermal conductivity values for the rigid PU foams synthesized with the recovered polyol without nanosilica and with fresh and recovered nanosilica.

It can be observed that, up to replacements of 60 wt %, the effective conductivity remains practically constant in a value between 0.060 and 0.065 W/m·K. Thus, there was no significant effect of the use of recovered polyol over this property. The typical effective conductivity range of commercial rigid PU foams is from 0.025 to 0.04 W/m·K [39]. The higher k observed in our case is due to the presence of the filler since the obtained value for the foam without nanosilica was of about 0.04 W/m·K, and the thermal conductivity of this filler is around 1.45 W/m·K [40]. Hence, using nanosilica as a filler worsens the insulation properties of the foam, because of the higher thermal conductivity of the filler. However, considering the thermal conductivity of the foam without nanosilica and the fact that the recovered polyol content up to a 60 wt % did not affect to the k value, it could be stated that it could be applied for the synthesis of RPU foams applied in the isolation sector up to this content.

Furthermore, it has been proved that changing the fresh nanosilica by the recovered one in the polyurethanes synthesis had no significant influence on the value of thermal conductivity.

#### 3.3.4. PU Foams Structural Characterization

FTIR analysis allowed us to verify the formation of the urethane linkages, as a result of the chemical reaction between the isocyanate and alcohol groups of the PMDI and rigid polyol, respectively. Besides, the FTIR allowed us to detect the presence of the nanosilica. FTIR spectra of the PU foams synthesized employing fresh and recovered polyol and nanosilica are presented in Figure 10.

As can be seen in Figure 10, all the foams presented very similar spectra. The presence of the secondary amine groups in the PU foam can be confirmed due to the N-H stretching vibration at a wavenumber of 3345–3380 cm^−1^ [41]. The complete reaction of isocyanate groups with the alcohol groups of polyol and the blowing agent is confirmed, due to the absence of the characteristic signal of isocyanate group (NCO) at 2276 cm^−1^, but for the case of the foams that contain a 45 and 60 wt % of recovered polyol, which would indicate an excess of isocyanate used in the synthesis of these foams [13,42]. Besides, the stretching vibration of C-H due to the CH_2_ group is found at 2970 and 2868 cm^−1^ [41], and the urea linkages generated during the reaction between PMDI and water, forming CO_2_, is located at 1641 cm^−1^, and corresponding to stretching vibration of urethane bond C = O [25]. The absorption at 1223 and 1094 cm^−1^ is attributed to C-O-C vibration of ether group in the PU foam [41,42]. Finally, at 1050 and 808 cm^−1^ are found in the Si-O-Si signal related to the filler presence in the rigid PU foams [14,29].

Figure 11 shows the microphotographs obtained by SEM with magnification ×100 of the rigid foams synthesized by means of employing several proportions of recovered polyols and with recovered nanosilica. Besides, it is also shown the appearance of the foam with 0.0 wt % of recovered polyol with fresh nanosilica and without it. 

It can be observed that the PU foams preserved the polyhedral cell structure up to a 30 wt % of recovered polyol, while for higher replacement percentages they presented irregular structure. However, foams with up to 60 wt % of recovered polyol kept closed cells what explain that the thermal conductivity values stay practically constant.

Moreover, with the addition of up to a 30 wt % of recovered polyol, it can be appreciated a reduction in the cell size. In the cases of 45 and 60 wt % of recovered polyol, a regular size of the cells was not observed, but some smaller ones could be appreciated. The average cell sizes for replacements of 0, 15 and 30 wt % of raw polyol were measured, resulting in 210, 180 and 150 μm, respectively.

The reduction of the cell size can be explained attending to the increase of the polyol mixture viscosity that hampers the CO_2_ expandability [34]. The higher content of recovered polyol, the higher the viscosity of the polyol mixture (Table 6) and, thus, the more restricted the CO_2_ to be expanded, giving as a result a decreasing in the PU foam cell size [32,33,34]. This also agrees with the density increase observed also up to a 30 wt % of recovered polyol.

On the other hand, polyurethane foams with fresh nanosilica and without it were analyzed (Figure 11b,c), measuring average cell sizes of 218 and 202 μm, respectively. The nanosilica did not influence on the cell structure or the average cell sizes. Besides, the addition of nanosilica should reduce the foam cell size since it can act as a nucleating agent, however, this is not the case due to the small amount of filler included in the foam syntheses.

It is important to note that all the rigid PU foams synthesized in this investigation were synthesized using the same recipe, changing exclusively the amount of isocyanate depending on the hydroxyl number of the polyol mixture used. However, the contents of water, surfactant and catalyst have been kept constant to adapt the recovered polyols to an industrial production of rigid PU foams. However, all the structural, physical and morphological properties of the foams synthesized could probably keep constant with a slight modification of the foam recipe, even for high recovered polyol values.

### 3.4. Recovery of the Glycolysis Agent

In order to improve the economy of the glycolysis process, the recovery of the glycol (crude glycerol) is going to be studied. Due to the excess that is used to obtain a biphasic product, it is especially interesting to carry out this recovery in order to ensure the economic feasibility of the process at industrial scale.

The recovery of the crude glycerol was carried out by means of a short path distillation, operating at a temperature of 220 °C and a vacuum of 25 mbar.

Figure 12 shows the GPC analysis of the bottom phase and the raffinate and extract obtained after distillation.

These results demonstrated the viability of recovering the crude glycerol employed in the glycolysis process, since the extract was composed mainly of crude glycerol with an approximate concentration of 99 wt %. On the other hand, the raffinate was composed of recovered polyol and reaction byproducts, which were present in the bottom phase.

Furthermore, to complete the characterization of the recovered crude glycerol, FTIR analysis were performed. These analyses are presented in Figure 13 together with the FTIR of raw crude glycerol.

Both spectra, that from the raw crude glycerol and that from the recovered one, were identical. Thus, these FTIR results confirm the high quality of the recovered crude glycerol, so it could be used in a new glycolysis process [43].

Finally, the recovery yield of the crude glycerol was estimated, which was 55%. Table 9 presents the data and calculations to obtain this result.

## 4. Conclusions

The extension of the glycolysis process to rigid polyurethane foams containing nanosilica as a filler is feasible using both, diethylene glycol and crude glycerol as glycolysis agents. However, diethylene glycol did not allow one to obtain a split phase product, which reduces their further applications. The use of crude glycerol as the glycolysis agent allowed one to obtain a split-phase product, driving to a recovered polyol with a 76.5 wt % of purity. Besides, the reaction conditions employed are suitable for the glycolysis of different types of flexible PU foam, which facilitates the use of this process in the general treatment of PU wastes. These process conditions were a temperature of 190 °C, a mass ratio of PU scraps to crude glycerol of 1:1.5 and tin octoate as a catalyst in a 2.3 wt % concentration in the glycolysis agent. In addition, the nanosilica used as a filler in the glycolyzed foams water was obtained with identical purity to fresh nanosilica.

The recovered polyol presented hydroxyl number of 854 mg KOH/g, molecular weight of 509 g/mol and functionality of 7.75, which are in the range of commercial polyols and was successfully applied for the replacement of up to a 60 wt % of a commercial rigid polyether polyol in the synthesis of new rigid PU foams. Recovered nanosilica was also used as a filler in the synthesized PU foams, in the same concentration that was present in the foams glycolyzed. In general, the synthesized foams exhibited regular structure without internal defects up to a 30 wt % of recovered polyol, changing to a more irregular cell shape for higher contents. A reduction in the cell sizes up to a 30 wt % content of recovered polyol is noted, probably due to an increase of the polyol mixture viscosity, which can restrict the CO_2_ expansion. This agreed with the variation trend observed for the density, which increased up to the 30 wt % and then kept constant up to 60 wt %.

On other hand, the maximum compressive strength remained constant with the incorporation of up to 60 wt % of recovered polyol and the Young’s modulus stayed also constant up to the 30 wt % with a reduction up to 28% for the recovered polyol content of 60 wt %. The mechanical properties were independent of using recovered or fresh nanosilica. Moreover, the improvement of the mechanical properties was demonstrated when the nanosilica was employed as a filler in the polyurethane foams since a foam without nanosilica was tested showing a σ_max_ and E values of 469.78 kPa and 11.60 MPa, respectively, which were a 11.5% and 16.9% lower than when adding the nanosilica as a filler.

In terms of thermal properties, replacements of raw polyol up to a 60 wt % did not affect significantly to the thermal conductivity of the foams, confirming the good foaming and structure of the foam with the recovered polyol. Thus, the recovered polyol can be used for the synthesis of insulating RPU foams. However, the use of nanosilica as a filler worsen the isolation properties because of the higher thermal conductivity of the filler (1.45 W/m·K) in comparison to the polyurethane foam (0.04 W/m·K).

Finally, the viability of recovering the excess glycolysis agent (GC) was studied and demonstrated. An extract or recovered crude glycerol was obtained with an approximate concentration of 99 wt % and its recovery yield was of 54.9%. The use of the recovered crude glycerol in new glycolysis processes would improve the economic feasibility of the treatment of these wastes at the industrial scale.

## Data Availability

MDPI Research Data Policies.

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
