# Peer review of "Glycolysis of Polyurethanes Composites Containing Nanosilica"

_polymers, 2021, doi:10.3390/polym13091418_

Round 1

Reviewer 1 Report

  1. The novelty of the work is not well emphasized.
  2. Besides, the paper is very carelessly prepared. The paper must be thoroughly proofread for lexical and grammatical mistakes.
  3. Abstract is unnecessarily long (266 words) and keywords, authors contribution, funding, conflict statement, and data availability statement are missing.
  4. Literature review is not properly conducted and very few papers from the recent years have been discussed. Only one paper from 2020 has been cited.
  5. The loThe characterization of Nano silica must be carried out and its important characteristics must be included (TEM and SAED is suggested).
  6. The SEM images are very poor in resolution. High magnification images with legible scale bars should be provide.
  7. The most critical issue is that the paper has 30% similarity with the previously published papers (Turnitin report attached).

Reviewer 2 Report

  • References: 15 from 32 references are self-citations. What is the novelty of presented work in comparison to other your works? It should be clearly stated.
  • Introduction should completed by comprehensive review (including description of
  • Title: In my opinion title should be "Glycolysis and glycerolysis..."
  • Sections: Keywords, Author contributions, Funding, Data Availability Statement, Conflict of interests - should be completed
  • Figures 3 and 4 should be combined
  • Hydroxyl values should be supported by standard deviation
  • Could you propose the reaction scheme for the performed glycolysis/glycerolysis?
  • Please, present the details about the polyol used for the synthesis of rigid polyurethane foams prepared for the glycolysis process. The details about chemical composition of glycolyzed foams should be provided
  • In my opinion, general scheme of the performed experiments with some technical details should be preapred (including following steps: synthesis of polyurethane foams with nanosilica-> glycolysis of prepared foams - > synthesis of PU foams using recovered polyol and nanosilica. All stages should be characterized separately) 
  • Line 300: Numbering should be corrected
  • Figures 6-8 should be replaced by the suitable Tables (average values with standard deviation)
  • FTIR spectra of synthesized polyurethane foams should be presented and discussed
  • The all results (FTIR, GPC etc.) for the glycolysis (using DEG) and glycerolysis should be presented in the manuscript 
  • Manuscript should carefully checked by the Authors and corrected. English language should be improved 

Reviewer 3 Report

The paper can be accepted in the present form

Author Response

Thanks to the referee for his kind consideration of this paper

Round 2

Reviewer 1 Report

The paper is revised but the issues of lacking novelty and overlapping text still exists. Further, the nanomaterial characterization is not done properly. The paper lacks contribution to existing knowledge base as well as scientific appropriateness.

Reviewer 2 Report

Work was significantly improved and can be accepted after minor revisions:

  • The data presented in Figure 8 should be introduced in Table 7. The changes of thermal conductivity values are negligible (so their presentation in separate Figure is not necessary) 
  • The quality of Figure 2, 3 and 5 should be improved. Please, use the same software as in the case of Figures 4, 6, 12 or 13
  • The quality of SEM micrographs (Figure 10) should be improved (for example, the visibility of scale is poor)
  • Detailed reaction scheme of glycolysis and glycolysis should be provided. Please discuss the products of the reactions, when the mass excess of glycolytic agent is used. The analysis of GPC results (description of fractions characterized by different average molecular weights) will be helpful. 

Round 3

Reviewer 1 Report

The paper has 24% similarity with the following previously published paper (Turnitin report attached for editor's perusal). Complete paragraphs have been directly copied. I recommend the paper to be declined.

    1. https://doi.org/10.1016/j.cej.2018.05.158 

Other major comments are given below.

  1. The nano-silica characterization is scientifically not appropriate. Merely SEM and XRD are done which do not provide necessary information. Please note that TEM and SAED are required to determine the true properties.
  2. XRD peaks are not identified.
  3. The SEM images are very low in quality with poor magnification levels. The scale bars are illegible.
  4. Very few papers from the recent years are cited. The bibliography is inadequate.
  5. The discussion section is too weak.
